# Socioeconomic Disparities in Diet Vary According to Migration Status among Adolescents in Belgium

**DOI:** 10.3390/nu11040812

**Published:** 2019-04-10

**Authors:** Manon Rouche, Bart de Clercq, Thérésa Lebacq, Maxim Dierckens, Nathalie Moreau, Lucille Desbouys, Isabelle Godin, Katia Castetbon

**Affiliations:** 1Research Center in Epidemiology, Biostatistics and Clinical Research, School of Public Health, Université libre de Bruxelles (ULB), 1040 Brussels, Belgium; theresa.lebacq@ulb.ac.be (T.L.); nathalie.moreau@ulb.ac.be (N.M.); lucille.desbouys@ulb.ac.be (L.D.); katia.castetbon@ulb.ac.be (K.C.); 2Department of Public Health and Primary Care, Ghent University (UGent), 9000 Ghent, Belgium; b.declercq@ugent.be (B.d.C.); maxim.dierckens@ugent.be (M.D.); 3Service d’Information Promotion Education Santé (SIPES), School of Public Health, Université libre de Bruxelles (ULB), 1040 Brussels, Belgium; 4Research Centre in Social Approaches to Health, School of Public Health, Université libre de Bruxelles (ULB), 1040 Brussels, Belgium; isabelle.godin@ulb.ac.be

**Keywords:** migration status, dietary habits, food frequency questionnaire, socioeconomic disparities, adolescents

## Abstract

Little information concerning social disparities in adolescent dietary habits is currently available, especially regarding migration status. The aim of the present study was to estimate socioeconomic disparities in dietary habits of school adolescents from different migration backgrounds. In the 2014 cross-sectional “Health Behavior in School-Aged Children” survey in Belgium, food consumption was estimated using a self-administrated short food frequency questionnaire. In total, 19,172 school adolescents aged 10–19 years were included in analyses. Multilevel multiple binary and multinomial logistic regressions were performed, stratified by migration status (natives, 2nd- and 1st-generation immigrants). Overall, immigrants more frequently consumed both healthy and unhealthy foods. Indeed, 32.4% of 1st-generation immigrants, 26.5% of 2nd-generation immigrants, and 16.7% of natives consumed fish ≥two days a week. Compared to those having a high family affluence scale (FAS), adolescents with a low FAS were more likely to consume chips and fries ≥once a day (vs. <once a day: Natives aRRR = 1.39 (95%CI: 1.12–1.73); NS in immigrants). Immigrants at schools in Flanders were less likely than those in Brussels to consume sugar-sweetened beverages 2–6 days a week (vs. ≤once a week: Natives aRRR = 1.86 (95%CI: 1.32–2.62); 2nd-generation immigrants aRRR = 1.52 (1.11–2.09); NS in 1st-generation immigrants). The migration gradient observed here underlines a process of acculturation. Narrower socioeconomic disparities in immigrant dietary habits compared with natives suggest that such habits are primarily defined by culture of origin. Nutrition interventions should thus include cultural components of dietary habits.

## 1. Introduction

High consumption of foods such as chips and fries [1] and sugar-sweetened beverages (SSB) [2] might be associated with increased noncommunicable diseases (NCD); by contrast, adequate consumption of fruits, vegetables [2,3], fish [2], and dairy products [4] might reduce NCD and all-cause mortality. Furthermore, it has been estimated that up to two-thirds of NCD social inequalities may be explained by dietary disparities [5,6].

In addition, evidence indicates that dietary habits during adolescence may continue into adulthood [7,8,9]. To implement effective prevention of NCD throughout the lifetime, disparities in adolescent eating behavior warrant elucidation. However, information on this topic is scarce in Western countries. Although several studies have pointed out the association between dietary habits and socioeconomic status (SES) among adolescents [10,11], SES may not explain all observed variations.

Among other determinants of dietary habits, migration status may play a role [12,13]; however, published studies are rare, even among adults, and are often oriented towards a specific ethnic group [12,13]. Studies on health related to migration have revealed a mortality advantage in immigrants compared to natives, despite the lower SES of most immigrants [14]. This paradox could be explained by the “healthy-migrant effect”, i.e., positive self-selection, and an unhealthy return-migration effect, also known as the “salmon-bias hypothesis” [14]. However, these selection processes are subject to caution; “beneficial cultural and behavioral factors”, like dietary habits, may be the most plausible explanation for this paradox [14]. The immigrant health advantage may also tend to wear off with length of stay in the host country, mostly due to an acculturation process [15,16], wherein foreign individuals partially integrate behaviors and cultural aspects of the host population while maintaining their roots [17]. This has been highlighted in some adult dietary studies and suggests gradual adaptation to the natives’ eating habits according to years spent in the host country [18]. In adolescents, dietary habits varying according to migration status (including a gradient across migration generations) might also be expected, but such investigations have thus far been limited [19,20].

An effect modification between migration status and socioeconomic characteristics was emphasized in a previous study on adult self-rated health: A gradient was revealed across social classes in natives, which was not the case for immigrants from poor countries [21]. Similarly, in adolescents, a clear gradient throughout family affluence categories was observed for health-related quality of life in natives but not in immigrants [22]. Therefore, a possible effect modification of migration status on socioeconomic characteristics should be considered when evaluating disparities in dietary habits.

The aim of this study was to estimate socioeconomic disparities in dietary habits of school adolescents from different migration backgrounds. We first hypothesized that adolescent immigrants have healthier dietary habits than natives and that food consumption frequencies increase or decrease gradually according to the migration generation, related to an acculturation process. Secondly, we assumed that migration status might modify the association between socioeconomic characteristics and dietary habits in adolescents.

## 2. Materials and Methods

Research was carried out using data from the “Health Behavior in School-Aged Children” (HBSC) survey conducted in 2014 in Belgium. The cross-national HBSC survey takes place every four years in around 40 countries in Europe and North America under the aegis of the World Health Organization (WHO) Regional Office for Europe. Its goal is to produce comprehensive indicators supporting implementation of health prevention and promotion policies and interventions. Questionnaires are self-administrated in the classroom, and anonymity and data confidentiality are guaranteed [23].

Belgium has a regionalized administration in which wide demographic variations are observed, including those concerning migration. In 2014 in Belgium, this study was carried out separately in French- and Dutch-speaking schools covering the three regions, Wallonia, Flanders, and Brussels, with the latter including both French- and Dutch-speaking schools.

This survey was carried out according to guidelines articulated in the Declaration of Helsinki. For French-speaking schools, the survey was approved by school authorities. This protocol was not submitted to a medical ethics committee in view of the topics and methods used for data collection (Belgian law of May 7, 2004 and Advisory Committee on Bioethics of Belgium, opinion n°40, 12/2/2007). For Dutch-speaking schools, the study was approved by the ethics review committee of the University Hospital of Ghent (project EC/2013/1145). Following advice from school authorities, no written consent was requested for French-speaking schools; for the Dutch-speaking schools, consent was passive. Adolescents were clearly informed about survey content and about their full right to refuse to fill out the questionnaire or answer specific questions. All procedures used during data collection enabled confidentiality and anonymity.

### 2.1. Sampling

The French- and Dutch-speaking surveys were conducted on a random sample stratified proportionally with the school networks and included public and private schools. In addition, in the French-speaking part, the sample was stratified proportionally with the province (*n* = 6); in the Dutch-speaking part, it was stratified proportionally with the form of education (ordinary, general, technical, vocational, art secondary education, and non-native newcomer classes).

In all regions, schools were first randomly selected based on an official list. Next, classes from fifth grade elementary school (corresponding to adolescents aged ± 10 years) to the final grade of the secondary school (corresponding to adolescents aged ± 18 years) were selected in each grade among the schools that agreed to participate in the study. All adolescents from selected classes were invited to participate on a voluntary basis. In French-speaking schools, classes were randomly selected. In Dutch-speaking schools, distributions of gender, grade, and form of education from the previous survey were used as temporary proxies to select classes.

In 2014, 781 schools in the French-speaking schools and 208 schools in the Dutch-speaking schools were invited to participate. Among these schools, 168 in the French- and 98 in the Dutch-speaking areas actually participated, corresponding, respectively, to a participation rate of 21.5% and 47.1%.

In total, 23,552 questionnaires were collected (Figure 1). Since, in Dutch-speaking schools, adolescents aged 20 or over were not questioned, only adolescents 10 to 19 years old were included in the joint database (*n* = 23,031). The basis sample included all participants who responded to all covariates and to the food consumption variable that was most frequently filled in, i.e., fruits. Thus, the maximum number of adolescents included in the analyses was 19,172 (Figure 1). For food consumption other than fruits, the sample size was slightly lower.

### 2.2. Measures

Food Frequency Questionnaire. Food data were collected using a validated short food frequency questionnaire (FFQ) [24,25] that included a total of 22 food groups (17 in the Dutch part, 18 in the French part, including 13 in common). Seven answer categories were proposed: “more than once a day”; “once a day”; “5–6 days a week”; “2–4 days a week”; “once a week”; “less than once a week”; and “never”.

Migration status. Adolescents born abroad and whose parents were not both born in Belgium were considered “1st-generation immigrants”. Adolescents born in Belgium and who had at least one parent born abroad were considered “2nd-generation immigrants”. Adolescents whose parents were born in Belgium were considered to be “natives”.

Geographical area of origin. Based on countries of origin of adolescents for 1st-generation immigrants and of parents for 2nd-generation immigrants, five categories were defined for complementary analyses: (1) Europe; (2) America; (3) Asia; (4) Middle East and North Africa; (5) Sub-Saharan Africa. For 2nd-generation immigrants, in the particular case where the parents were from two different geographical areas, the category was randomly chosen between the mother’s and the father’s area of origin.

Family affluence. The family affluence scale (FAS) is composed of six items and has been validated in Europe [26]. The FAS score ranged from 0 to 13 and was divided into three categories after ridit analysis transformation—“low”, “medium”, and “high”—corresponding, in the national sample, to the 20% of adolescents with the lowest FAS scores, the 60% of adolescents with the intermediate score, and the 20% of adolescents with the highest FAS scores, respectively.

Parental working status. Based on parental employment and reason for parental unemployment, four categories were defined: (1) Adolescents with both parents working; (2) adolescents with parent(s) not working (those with a parent not working and without a second parent were placed in this category (*n* = 365)); (3) adolescents with one working parent and the other at home (housewife/husband, pre-retired, disabled or student); (4) adolescents with one working parent and the other “absent” from home (seeking a job, or adolescent not living with the second parent).

In addition, gender, age, family structure, siblings, and school region were taken into account in the analyses.

### 2.3. Statistical Analyses

#### 2.3.1. Reprocessing Data

For all food items, categorization was defined so as to correspond as closely as possible to Belgian nutritional policies [27] but was also determined by the original answer modalities. Food consumption was first divided into three categories: A category corresponding as closely as possible to the Belgian nutritional policies, a category further removed from Belgian nutritional policies, and an intermediate category. If the intermediate category did not provide additional information, categorization was reduced to two categories. Frequency of *fruit and vegetable consumption* was classified into three categories: “>once a day”, “5–7 days a week” (low frequency), and “<5 days a week” (very low frequency). Frequency of *fish consumption* was classified into: “≥two days a week” and “<two days a week” (low frequency). Milk (whole and semi-skimmed/skimmed), cheese, and other dairy product frequencies were transformed into consumption per month of 30 days and added up to obtain frequency of *dairy consumption*; this was then divided into total consumption of 31 days or more, which corresponded to consumption “>once a day”, or else to consumption “≤once a day” (low frequency). Consumption of chips and fries and SSB was similarly processed. Frequency of *chips and fries* consumption was classified into: “<once a day”, strictly corresponding to consumption under 25 days, and “≥once a day” (high frequency). Frequency of *SSB consumption* was classified into: “≤once a week”, corresponding to total consumption under 5 days, “2–6 days a week” (high frequency) and “≥once a day” (very high frequency), corresponding to a total of 25 days or more.

#### 2.3.2. Modeling

Due to a significant effect modification of migration status on several covariates for each food group analyzed, analyses were stratified by migration status. Since individuals were nested within schools, multilevel models were used. In each model for each food group, the school effect was controlled and estimated. The “null” model referred to the estimation of the school effect. Associations between covariates and a given food consumption were then evaluated by univariate regression, corresponding to “model 1”. Multilevel binary logistic regression was performed for food groups with consumption frequencies in two categories (odds ratio (OR) and 95% CI); multilevel multinomial logistic regression was used for food groups with consumption frequencies falling into three categories (relative risk ratio (RRR) and 95%CI). The reference category was assumed to be the most favorable category in terms of health. All covariates with a *p* value < 0.20 were included in the initial multivariable regression model. Manual backward stepwise selection was used to determine the final model: It consisted of iteratively removing the predictor with the highest *p* value, and higher than 0.05. Following removal, confounding was evaluated with a tolerance threshold for variation of OR and RRR set at 10%. If variation was greater than 10%, the variable was then retained in the model. In order to facilitate comparisons between migration strata, a predictor significantly associated with food consumption in a given migration status group was retained for all other migration status groups. The results of the regressions are graphically presented and available as supplementary tables (Appendix A).

In order to estimate the role of the immigrants’ geographical area of origin on socioeconomic and sociodemographic disparities, the same modeling was carried out only in immigrant adolescents and adjusted for the geographical area of origin.

Colinearity and fitting of models were verified. Statistical significance of tests was set at 0.05. All analyses were performed using Stata/IC 14^®^ (StataCorp, College Station, TX, USA).

## 3. Results

The sample of 19,172 school participants included 69.6% natives, 22.0% 2nd-generation immigrants, and 8.4% 1st-generation immigrants. In univariate analyses, differences in sociodemographic and socioeconomic characteristics were observed according to migration status (Table 1).

Irrespective of migration status, one adolescent out of five ate fruits (17.2%) or vegetables (18.8%) >once a day and fish ≥two days a week (20.2%) (data not tabulated). Eight pupils out of ten (79.6%) ate dairy products >once a day. Nearly half of the adolescents (43.7%) drank SSB ≥once a day; fewer than one-eighth (12.2%) ate chips and fries ≥once a day (data not tabulated).

Immigrants significantly more often ate fruits (>once a day and 5–7 days a week), vegetables (>once a day) and fish (≥two days a week) than did natives (Figure 2). In addition, immigrants significantly more often consumed SSB (≥once a day), and chips and fries (≥once a day) than natives. Moreover, the proportion of 2nd-generation immigrants having low or very low intake of vegetables, fish, chips and fries was significantly at an intermediate level, i.e., higher (or lower) than that of natives, and lower (or higher) than that of 1st-generation immigrants. However, a significant difference in dairy product consumption was observed only between 2nd-generation immigrants and natives (Figure 2).

In immigrants only, and compared with immigrants from a European country, statistically significant differences in food consumption frequencies were found in those from the Middle East and North Africa areas, and from sub-Saharan Africa, for vegetable, fish, dairy, chips and fries, and SSB (Appendix A). Immigrants from America also slightly differed regarding SSB and chips and fries consumption, and those from Asia for chips and fries consumption. Overall, estimates of the SES characteristics were not modified by the addition of origins in the modeling except only for siblings in 1st-generation immigrants for chips and fries consumption (data not shown). 

### 3.1. Fruit Consumption (Reference Category: >Once a Day)

In all migration strata, the likelihood of very low fruit consumption frequency (<5 days a week) significantly decreased with the FAS (except in 1st-generation immigrants: NS for medium FAS vs. high FAS) (Figure 3). In addition, natives having a medium FAS were more likely to eat fruits 5–7 days a week (low frequency) than those with a high FAS. Native adolescents in blended families were more likely to declare very low or low fruit consumption compared to adolescents with two parents; native adolescents in single-parent families also had greater odds of very low frequency. There was no significant association with parental working status. In all migration strata, adolescents in Flanders were significantly more likely to declare low or very low fruit consumption than adolescents in Brussels-Capital (Figure 3).

Moreover, age was significantly associated with fruit consumption frequency in all migration strata (Figure 3). This was also the case for gender in natives and 2nd-generation immigrants, whereas the existence of siblings was associated with fruit consumption frequency only in natives. The school effect upon the very low fruit consumption of all migration groups was significant, as was its effect upon the low frequency found in natives (Figure 3).

### 3.2. Vegetable Consumption (Reference Category: >Once a Day)

In natives and 1st-generation immigrants only, the odds of very low vegetable consumption frequency (<5 days a week) decreased with FAS (Figure 4). The same applied to low frequency (5–7 days a week) in natives. In 2nd-generation immigrants, adolescents having a medium FAS were more likely to declare very low or low vegetable consumption than those with a high FAS; in addition, adolescents having a low FAS were more likely to eat vegetables <5 days a week.

Compared with adolescents with two parents, native adolescents from a single-parent family were at greater odds of consuming vegetables <5 days a week (Figure 4). Native adolescents with two working parents were more likely to declare a low frequency than adolescents with one parent who worked and the other who stayed at home. Moreover, 2nd-generation immigrants with both working parents were less likely to declare a very low frequency. Compared with adolescents in Brussels-Capital, adolescents in Flanders were more likely to eat vegetables 5–7 days a week in all migration strata; in addition, immigrants in Flanders were more likely to declare a very low frequency, whereas 2nd-generation immigrants in Wallonia were less likely to do so (Figure 4).

Gender was significantly associated with vegetable consumption frequency in all migration strata. This was also the case for age in natives and 2nd-generation immigrants, while the existence of siblings was associated only in natives. The school effect was significant only in natives (Figure 4).

### 3.3. Fish Consumption (Reference Category: ≥Two Days a Week)

In natives and 2nd-generation immigrants, the likelihood of eating fish at low frequency (<two days a week) decreased with FAS. Moreover, low fish consumption was more frequent in adolescents from blended families than in those having two parents (Figure 5). Compared with adolescents with one working parent and the other who stayed at home, 2nd-generation immigrants with no working parents were less likely to eat fish at low frequency. In 1st-generation immigrants, adolescents whose parents both worked were more likely to declare low fish-eating frequency. For all migration strata, compared with Brussels-Capital, adolescents in Flanders and Wallonia were significantly more likely to declare low frequency (Figure 5).

In addition, gender was significantly associated with fish consumption frequency in natives and 2nd-generation immigrants. The school effect was significant in natives and 2nd-generation immigrants.

### 3.4. Dairy Product Consumption (Reference Category: >Once a Day)

For all migration situations, neither FAS nor family structure nor parental working status was significantly associated with dairy consumption (Figure 6). Compared with Brussels-Capital, natives and 2nd-generation immigrants in Flanders were at significantly lower odds of consuming dairy foods ≤once a day (Figure 6). 

In addition, dairy consumption frequency was significantly associated with gender in all migration strata and with age in natives and 2nd-generation immigrants, whereas the presence of siblings was significant only in natives. The school effect was significant in all migration strata (Figure 6).

### 3.5. Chips and Fries Consumption (Reference Category: <Once a Day)

Compared with those having a high FAS, natives with a low FAS were significantly more likely to declare frequent eating of chips and fries (≥once a day) (Figure 7). High frequency was also more likely in natives from blended or single-parent families than in those with two parents. Compared with adolescents having one working parent and the other who stayed at home, adolescents with two working parents were less likely to consume chips and fries ≥once a day whatever their migration status. In natives and 2nd-generation immigrants with no working parents, high frequency was more likely. In Flanders, compared to Brussels-Capital, adolescents among all migration groups were significantly less likely to declare high consumption of chips and fries; in Wallonia, this was also the case for 2nd- and 1st-generation immigrants (Figure 7).

In addition, frequent eating of chips and fries was significantly associated with gender in all migration strata, and with age in natives and 2nd-generation immigrants. Sibling presence was significantly associated only in 1st-generation immigrants. The school effect was significant in all migration strata (Figure 7).

### 3.6. Sugar-Sweetened Beverages (Reference Category: ≤Once a Week)

In natives and 2nd-generation immigrants, the odds of declaring very high SSB consumption (≥once a day) decreased with FAS (Figure 8). For all migration strata, adolescents from blended families were significantly more likely to declare very high frequency. In addition, natives from blended families were more likely to consume SSB at high frequency (2–6 days a week). In natives and 2nd-generation immigrants, very high SSB frequency was more often seen in adolescents from single-parent families (vs. two-parent families) and in adolescents with nonworking parents (vs. one working parent and the other at home) and was less likely in adolescents with both parents working. In 2nd-generation immigrants only, very high SSB frequency was also less likely in adolescents with one parent working and the other absent from the home. In Flanders, compared with Brussels-Capital, natives and 2nd-generation immigrants were more likely to consume SSB 2–6 days a week.

In addition, “gender and age” was significantly associated with SSB frequency in all migration strata, whereas “siblings” was significant only in natives. The school effect was significant only in natives (Figure 8).

## 4. Discussion

The aim of the present study was to estimate socioeconomic disparities in dietary habits of school adolescents in Belgium from different migration backgrounds. Our results emphasize that the migration component that was rarely considered in previous studies is fundamental regarding dietary behavior at these ages. Indeed, dietary habits differed according to migration strata. Furthermore, socioeconomic disparities varied amongst the migration groups: For all food groups, disparities were particularly wide in natives and more limited in 1st-generation immigrants. Overall, the sociodemographic and socioeconomic disparities observed in immigrants did not change after adjusting for their geographical area of origin. By food group, the widest socioeconomic and cultural disparities were observed for SSB and vegetables, and the least for dairy foods. Such findings provide interesting and original hypotheses that could further support the development of health promotion interventions in the future.

### 4.1. Dietary Acculturation

In descriptive analyses, immigrant adolescents, whether of the 1st or 2nd generation, were more likely to frequently consume both healthy (fruits, vegetables, and fish, but not dairy products) and unhealthy foods (chips and fries, SSB). In addition, a migration gradient in food frequencies was underlined for vegetables, fish, and chips and fries: Consumption gradually increased (for healthy food) or decreased (for unhealthy food) from natives to 2nd-generation immigrants, and from 2nd-generation immigrants to 1st-generation immigrants. However, no significant differences were found between 1st- and 2nd-generation immigrants regarding consumption of fruits, dairy products, and SSB. 

The situation of 2nd-generation immigrants in terms of food consumption, intermediate between natives and 1st-generation immigrants, suggests ongoing acculturation. The interplay of host behavior and culture with that of immigrants may lead to a mixture of healthy and unhealthy dietary habits. Indeed, at a given age, 2nd-generation immigrants have probably been in Belgium much longer than 1st-generation immigrants and are therefore more likely to be further engaged in the process of integration of culinary habits of the host country and partial substitution of family roots, as reported for adults in different countries [18]. In addition to the migration generation, the region of origin may play differently in the acquisition of European dietary habits for immigrants. Indeed, differences in dietary habits were of lower size between European, American, and Asian, than between European and African and Middle-East immigrants. Nevertheless, the results could not be precisely interpreted due to the cultural heterogeneity remaining in this categorization by the geographical area of origin.

The acculturation process may depend on factors such as accessibility and affordability, acting as an “external push”, and on individual factors such as curiosity, acting as an “internal pull” [28], which might encourage the adding of novel foods to the traditional diet, thus offering wider diversity. Such diversity could result in greater intake via gradual adaptation to new food products [29]. This may also explain why immigrants more frequently ate almost each food group studied. Maintaining traditional food habits implies the availability and accessibility of such food; when this is not the case, then people of foreign culture might progressively abandon their diet in favor of the host diet [30]. The acculturation process should also be further studied by considering the age of arrival in Belgium of the 1st-generation immigrants, unavailable in this survey.

### 4.2. Socioeconomic Disparities in Dietary Habits

Our results emphasize several socioeconomic disparities in dietary habits in adolescents, mainly in natives and 2nd-generation immigrants. Indeed, adolescents with lower FAS less frequently ate healthy foods like fruits and more frequently consumed unhealthy foods like SSB, consistent with previous studies [10,11]. These disparities may be explained by a lower level of familiarity or adoption of dietary recommendations by parents [31], and by the affordability of healthy foods [32].

Some disparities related to family structure were also revealed, mainly in natives and 2nd-generation immigrants. In line with previous studies [33,34], adolescents from blended or single-parent families more frequently ate chips and fries and SSB and less frequently ate fruits, vegetables and fish. Single-parent families often have fewer financial resources, thereby impairing their access to healthy foods [32]. Indeed, in our sample, 30.8% of adolescents belonging to a single-parent family were in a low FAS compared to 14.0% of adolescents from two-parent families and 15.7% of adolescents from blended families (data not shown). In addition, single parents may also have less time for monitoring meals compared to dual-parent families [33,35]. Adolescents from blended families also tended to have less healthy food habits than adolescents from two-parent families. Indeed, stepparents may have fewer opportunities for active involvement in their stepchildren’s education and health [33].

Moreover, parental work status disparities were observed, mainly in natives and 2nd-generation immigrants. In our study, parental working status was mainly related to the socioeconomic condition: 10.2% of adolescents with both parents working had a low FAS, while 51.8% of adolescents with no parents working fell into this category (*p* < 0.001; 24.1% in those with one parent working and one at home, 31.5% for one parent working and the other “absent” from home). After adjustment for FAS and other covariates, our results related to parental working status were mixed. Indeed, compared to adolescents with both parents working, those with no working parents were more likely to frequently eat vegetables and fish, but they also more frequently ate chips and fries and SSB (data not shown). These results might appear surprising if we assume that parental working status is related only to socioeconomic conditions. However, they suggest an interplay between free time and work. Indeed, parental working status might also indicate that fully employed parents have less time to cook [36].

For all migration strata, eating habits varied according to the school region. Numerous differences were found in consumption frequencies between adolescents attending schools in Flanders, mainly Dutch-speaking, and those in Brussels and Wallonia, both primarily French-speaking. Another study indicating similar regional linguistic specificities in consumption of vegetables, dairy products, fish, and SSB in Switzerland hypothesized a possible influence of culture and eating habits of neighboring countries [37], which might also apply to Belgium: Culinary customs in Flanders may be strongly influenced by the Netherlands, while those in Brussels, although they do not border, and in Wallonia, may be influenced by France, which shares the same language, i.e., French. Further, several differences were pointed out between Brussels and Wallonia, mainly concerning 1st- and 2nd-generation immigrants. In 2011, nearly half of the inhabitants of the Brussels-Capital Region (42.4%) were born outside of Belgium, compared to 10.2% in Flanders and 14.1% in Wallonia [38]. Since the Brussels-Capital region is multicultural, this vast proportion of immigrants might contribute to slowing down the acculturation process; indeed, immigrants are usually surrounded by other immigrants [17]. By contrast, in Wallonia, such a process may have been accelerated, meaning that food habits in immigrants would differ from those of immigrants in Brussels.

Socioeconomic disparities were measurable in all food groups, except for dairy products, for which disparities were statistically significant only for the school region. The absence of socioeconomic disparities in consumption of dairy foods (milk, cheese, and other) could be explained by the diversity of these products and their overall affordability. 

### 4.3. Sociodemographic Disparities in Dietary Habits

Our results underlined gender disparities in food consumption in almost all migration strata. Compared to boys, girls were more likely to more frequently eat fruits and vegetables, which could be explained by taste preferences [39,40], health beliefs, and greater concern about weight [41].

After adjustment for other covariates, sibling disparities continued to be unfavorable to the single child (except for chips and fries in 1st-generation immigrants). The sibling role in food consumption might be explained by two opposite phenomena: “Modeling” leads to imitation of the model (i.e., of the sibling), whereas “de-identification” leads to differentiation from the sibling [42]. The absence of sibling disparities in immigrants could be explained by the manner in which society views sibling relationships and their respective roles [43]. Thus, immigrants in our study might come from countries that do not promote sibling relationships. However, the sibling role has rarely been examined in dietary studies irrespective of migration status [44], thus preventing interpretation. Certain psychologists have suggested that older siblings may influence health behavior [45]; thus, birth order should be considered when evaluating the association between diet and siblings.

### 4.4. Strengths and Limitations

Due to the cross-sectional design of the HBSC survey and use of self-administrated questionnaires, a substantial sample size was obtained in each region of Belgium, along with a wide range of topics addressed. Although both Dutch- and French-speaking surveys were conducted separately and in different languages, standardization of the questionnaires according to the international HBSC protocol made it possible to combine data sets [23]. However, the two surveys were not identical. For instance, they were differently stratified in order to reach the representativeness of the linguistic regions. Therefore, the generalization of the results to the school population in Belgium should be interpreted cautiously. A second point related to the independence of studies is the use of reverse order categorization for food frequencies (“never” to “>once a day” for the Dutch-speaking survey versus “>once a day” to “never” for the French-speaking survey) may have contributed to lower frequencies of fruit and vegetable consumption in Flanders compared to Wallonia, since initial responses may have been chosen more frequently. This discrepancy could explain why we obtained results differing from the final national food consumption survey (based on 24-h recall) [46]. The short HBSC FFQ might also lead to inaccuracies due to use of large food group names rather than exact food names, and to the overestimation of consumption frequencies [25]. However, it has been validated in Belgium through a comparison with a seven-day record [24,25]. The conclusion was that it can be considered reliable for “ranking subjects according to consumption of the individual food items” [25]. In addition, we can only conclude about frequencies and the results must be interpreted as such given that a more frequent consumption does not necessarily imply—nor rules out—a higher food amount or a higher energy intake. 

A significant strength of the current study was the use of multilevel analyses controlling for the school effect and, therefore, cluster bias. Nevertheless, further interpreting the school effect is difficult given that food-related school characteristics were unavailable in this study. To better understand such effects, further studies should simultaneously consider contextual characteristics of the schools, such as implementation of nutritional actions and canteen use. For some food groups (fruits, vegetables, and SSB), categorization was in three instead of two, as is the case in numerous studies using categorization and FFQ. Although disparities were narrower for the intermediate category and, therefore, little difference from the reference category was observed, intermediate categories provided new information in certain cases. Indeed, in natives and 2nd-generation immigrants, age-related disparities existed for vegetables 5–7 days a week but not for vegetables <5 days a week; in addition, school-region-related disparities existed for SSB consumption 2–6 days a week but not for SSB ≥once a day. Although difficult to interpret, the school effect for natives might be a protective factor for fruits 5–7 days a week and a risk factor for fruits <5 days a week.

Another limitation was the rather small sample size of 1st-generation immigrants (*n*_max_ = 1605), leading to fewer participants in some categories (*n*_min_ = 159) and resulting in loss of statistical power. However, confidence intervals of OR and RRR (Figure 3, Figure 4, Figure 5, Figure 6, Figure 7 and Figure 8) suggested that nonsignificant results in 1st-generation immigrants were mainly due to fewer disparities rather than a lack of precision. This rather small sample size was also restrictive for in-depth analysis stratified by countries or continental regions of origin. To get around the small sample, results were subsequently adjusted for geographical areas of origin. Nevertheless, some cultural heterogeneity remained in this categorization and did not help to precisely interpret the findings.

Several biases could also be highlighted. First, adolescents may have overreported consumption of healthy foods and underreported consumption of unhealthy products due to social pressure [47]. Second, we observed differential distribution of fish and SSB consumption and of several covariates (migration status, gender, family structure, FAS, parental working status, and school region) between included participants and eligible participants not included in analyses due to missing data (Appendix A), leading to selection bias. Interpretation of results should thus be approached with caution, although some differences in percentages were slight, and statistical significance was mainly due to the large sample size. The generalization of the results is limited to the school adolescents, especially for the eldest beyond the legal school age (18 years of age in Belgium). It should also be interpreted cautiously due to the relatively low participation rate of schools. 

## 5. Conclusions

Overall, rather poor adolescent dietary habits indicate that efforts should be made to improve knowledge and further prevent NCD in adulthood. The process of acculturation of dietary habits pointed out here warrants confirmation, taking into consideration the number of years in the host country and the age of arrival in that country. Narrower socioeconomic disparities in dietary habits among 2nd- and 1st-generation immigrants compared to natives suggest the prevailing role of culture in immigrant dietary habits with respect to socioeconomic conditions. Finally, our study reveals that interventions aimed at improving dietary habits in adolescents must take into account the cultural component of dietary habits, especially in immigrant adolescents. However, further research is needed to better understand the role of culture and its interaction with socioeconomic components in dietary habits.

## Figures and Tables

**Figure 1 nutrients-11-00812-f001:**
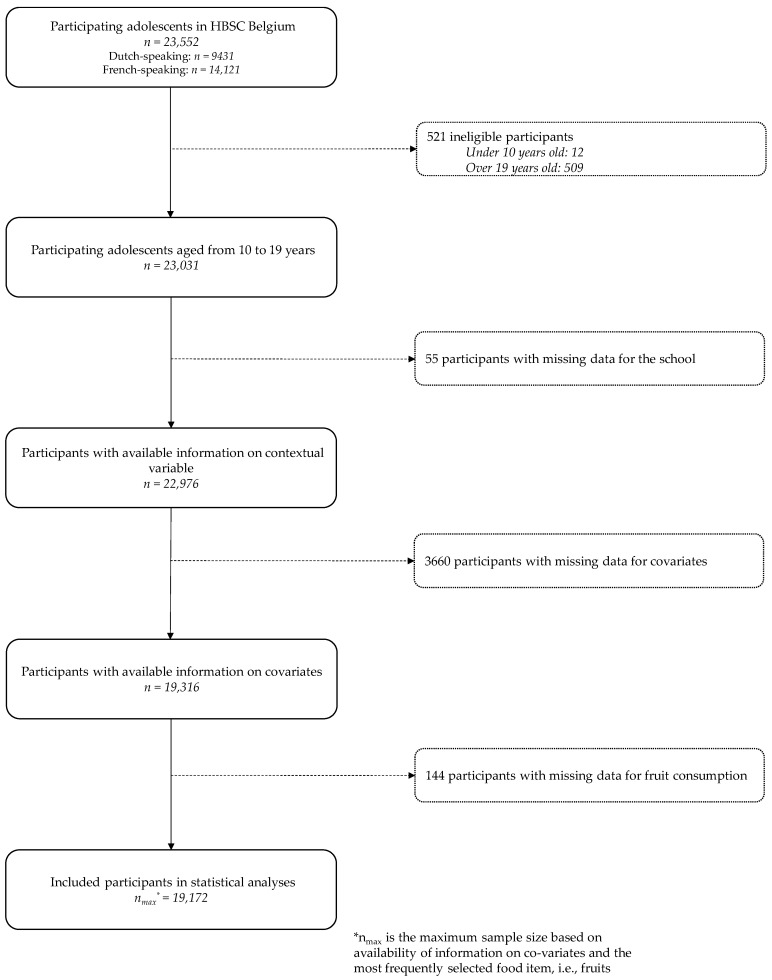
Inclusion diagram of adolescents in Belgium, “Health Behavior in School-Aged Children” (HBSC), 2014.

**Figure 2 nutrients-11-00812-f002:**
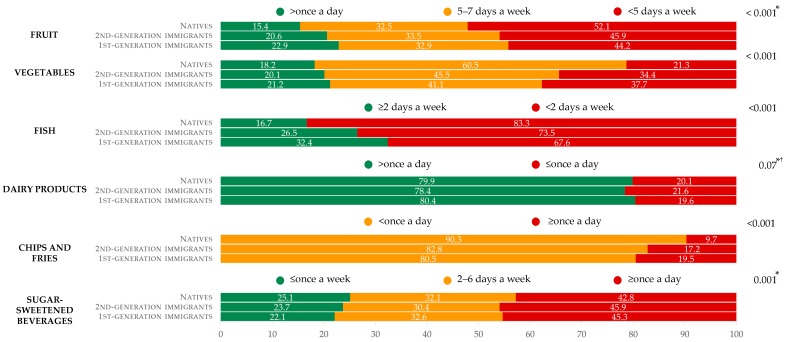
Distribution of food consumption frequencies by migration status in the sample, HBSC, Belgium, 2014. Favorable, moderately favorable, and unfavorable behaviors are represented, respectively, in green, orange, and red. * Nonsignificant difference between 2nd- and 1st-generation immigrants, † nonsignificant difference between natives and 1st-generation immigrants.

**Figure 3 nutrients-11-00812-f003:**
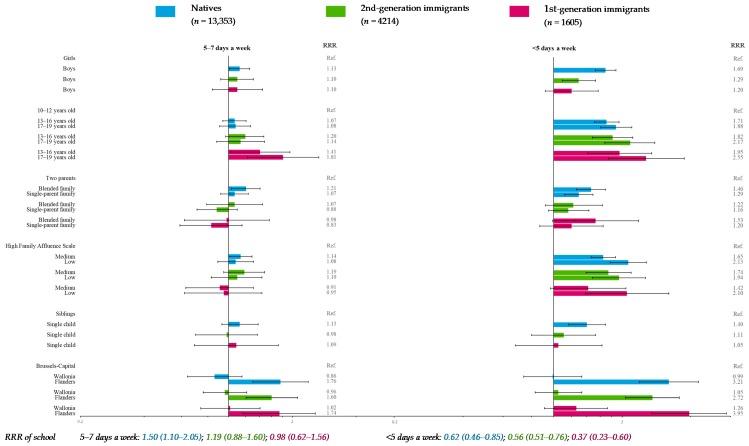
Multiple multilevel multinomial regression* for fruit consumption (reference category: >once a day) stratified by migration status—HBSC, Belgium, 2014 (*n* = 19,172). * RRR < 1: More favorable for health; RRR > 1: Less favorable for health.

**Figure 4 nutrients-11-00812-f004:**
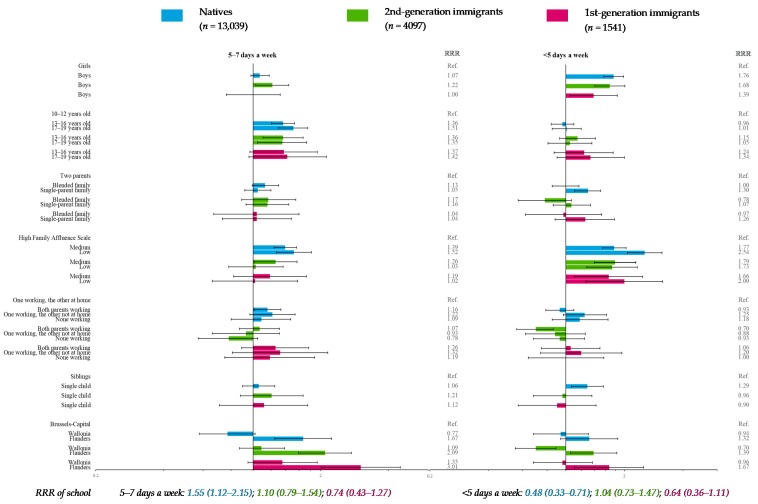
Multiple multilevel multinomial regression* for vegetable consumption (reference category: >once a day) stratified by migration status—HBSC, Belgium, 2014 (*n* = 18,974). * RRR < 1: More favorable for health; RRR > 1: Less favorable for health.

**Figure 5 nutrients-11-00812-f005:**
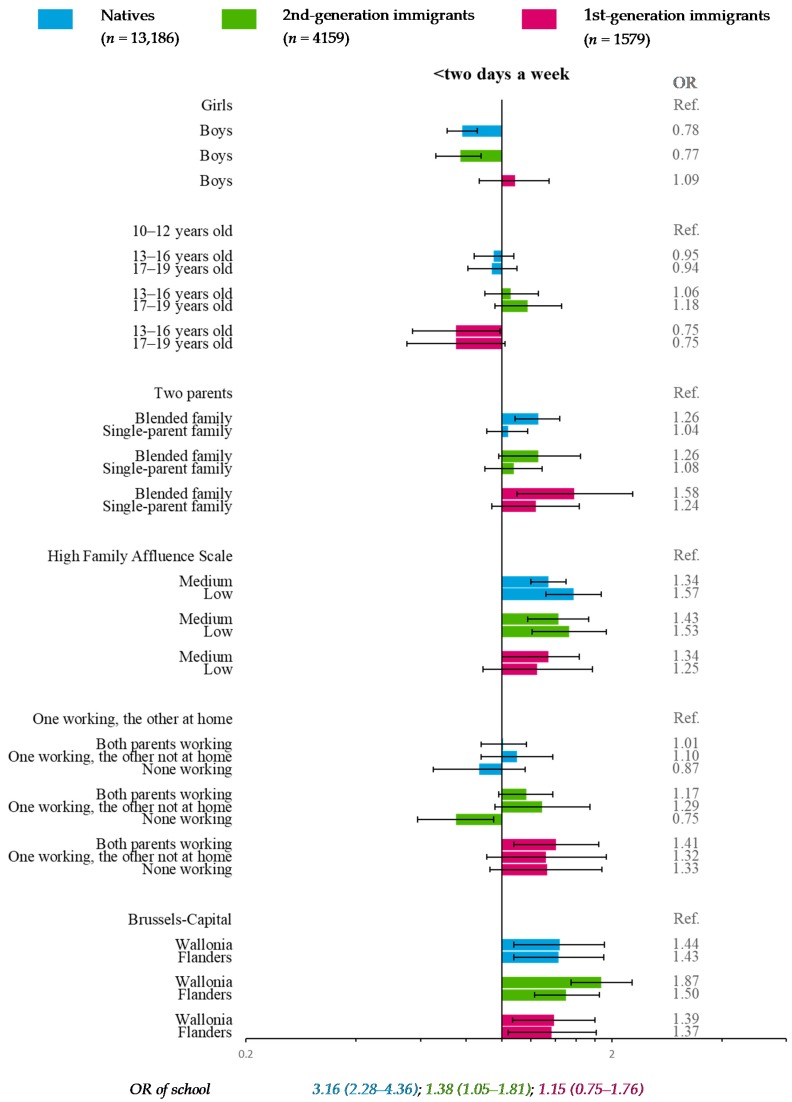
Multiple multilevel logistic regression* for fish consumption (reference category: ≥two days a week) stratified by migration status—HBSC, Belgium, 2014 (*n* = 18,924). * OR < 1: More favorable for health; OR > 1: Less favorable for health.

**Figure 6 nutrients-11-00812-f006:**
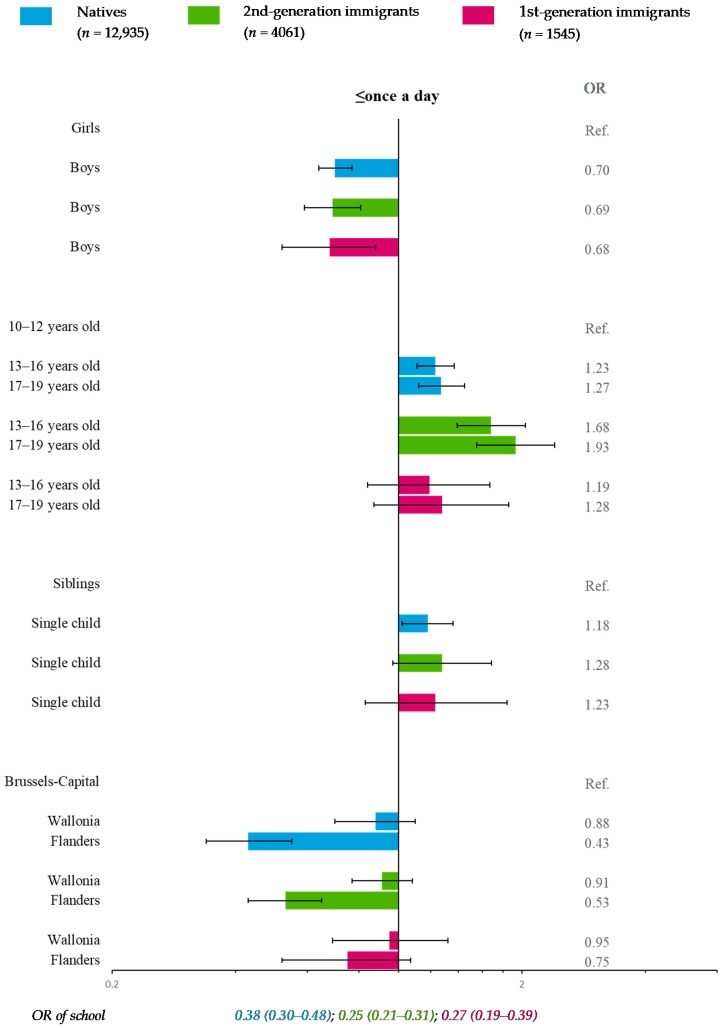
Multiple multilevel logistic regression* for dairy product consumption (reference category: >once a day) stratified by migration status—HBSC, Belgium, 2014 (*n* = 18,541). * OR < 1: More favorable for health; OR > 1: Less favorable for health.

**Figure 7 nutrients-11-00812-f007:**
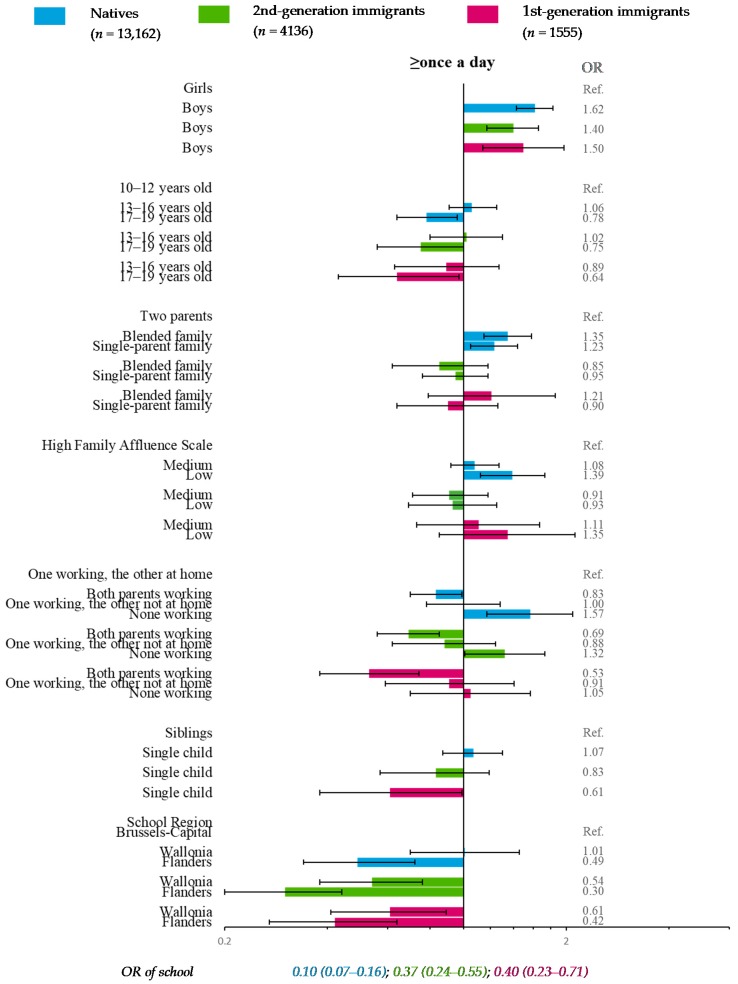
Multiple multilevel logistic regression* for chips and fries consumption (reference category: <once a day) stratified by migration status—HBSC, Belgium, 2014 (*n* = 18,853). * OR < 1: More favorable for health; OR > 1: Less favorable for health.

**Figure 8 nutrients-11-00812-f008:**
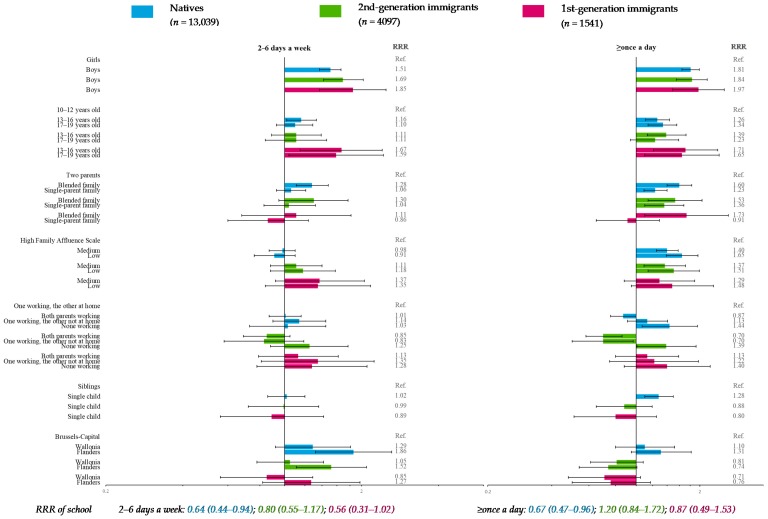
Multiple multilevel multinomial regression* for sugar-sweetened beverages consumption (reference category: ≤once a week) stratified by migration status—HBSC, Belgium, 2014 (*n* = 18,642). * RRR < 1: More favorable for health; RRR > 1: Less favorable for health.

**Table 1 nutrients-11-00812-t001:** Sociodemographic and socioeconomic characteristics of the sample overall and by migration status—HBSC, Belgium, 2014.

Variables	Sample	Natives	2nd-Generation Immigrants	1st-Generation Immigrants	*p* value
	(*n* = 19,172) %	(*n* = 13,353) %	(*n* = 4214) %	(*n* = 1605) %	
**Gender**					<0.001 ^†^
Boys	50.6	51.5	47.5	51.7	
Girls	49.4	48.5	52.5	48.3	
**Age**					<0.001 ^¥^
10–12 years	28.8	29.6	28.7	22.5	
13–16 years	50.2	49.9	50.6	51.4	
17–19 years	21.0	20.5	20.7	26.1	
**Family structure ^a^**					<0.001
Two parents	66.4	66.1	67.9	65.0	
Blended family	14.1	15.8	9.9	12.0	
Single-parent family	19.5	18.1	22.2	23.0	
**Family Affluence Scale ^a^**					<0.001
High	19.4	20.6	17.0	16.4	
Medium	63.7	66.1	59.8	53.6	
Low	16.9	13.3	23.2	30.0	
**Parental working status ^a^**					<0.001
Both parents working	68.4	76.1	51.9	47.0	
One working, the other at home	17.4	13.4	27.8	23.7	
One working, the other not at home	8.1	7.2	8.5	14.0	
None working	6.1	3.3	11.8	15.3	
**Siblings**					<0.001 ^†^
Single child	9.3	9.8	7.5	10.5	
Siblings	90.7	90.2	92.5	89.5	
**School Region**					<0.001
Brussels-Capital	11.4	3.4	29.2	31.7	
Wallonia	46.6	48.7	43.0	38.2	
Flanders	42.0	47.9	27.8	30.1	
**Geographical area of origin**					<0.001 *
Europe			43.7	59.8	
America			3.0	6.7	
Asia			7.3	7.7	
Middle East and South Africa			36.0	13.1	
Sub-Saharan Africa			10.0	12.7	

^a^ For details, see *Methods* section, ^†^ Nonsignificant difference between natives and 1st-generation immigrants, ^¥^ Nonsignificant difference between natives and 2nd-generation immigrants, * Comparison between 1st- and 2nd-generation immigrants.

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
