# Peer review of "Socioeconomic Disparities in Diet Vary According to Migration Status among Adolescents in Belgium"

_nutrients, 2019, doi:10.3390/nu11040812_

Round 1
Reviewer 1 Report
Line 70 - say "emphasized" or "outlined" instead of "underlined.
Line 93 - say "guidelines articulated in" instead of "laid down"
I thought the paper was very well done.
Author Response
Line 70 - say "emphasized" or "outlined" instead of "underlined.
We have replaced “underlined” by “emphasized” line 71.
Line 93 - say "guidelines articulated in" instead of "laid down"
We have replaced “laid down” by “articulated in” line 95.
Reviewer 2 Report
The authors present a very interesting and well-done study of a very important topic. I just have a few minor comments or concerns.
In the abstract please do not abbreviate FAS initially. I did not know what it meant until the body of the manuscript.
Results:
In Table 1. , the values in some way should be labeled as a percent
I found Figures 3-8 a little confusing. More explanation in the legend would be helpful. Perhaps marking a favorable or healthful direction?
I’m also confused about why there are no values for girls in the tables. Please explain or make this clearer.
I think there should be more discussion on why the results are important.
Author Response
In the abstract please do not abbreviate FAS initially. I did not know what it meant until the body of the manuscript.
We have replaced “FAS” by “Family Affluence Scale (FAS)” line 30.
Results:
In Table 1. , the values in some way should be labeled as a percent
We have made it clearer that values displayed in Table 1 were percent.
I found Figures 3-8 a little confusing. More explanation in the legend would be helpful. Perhaps marking a favorable or healthful direction?
We have added a footnote indicating how to interpret the direction in figures 3-8, lines 289, 293, 298, 341, 345, 349.
I’m also confused about why there are no values for girls in the tables. Please explain or make this clearer.
We have double-checked the tables and results for girls are actually presented. Maybe the misleading comes from the fact that given that “girls” was the reference category for gender, the value ‘1’ was reported for lines corresponding to ‘girls’ in tables S1-S6.
I think there should be more discussion on why the results are important.
We have modified the manuscript lines 351-354 and lines 359-361.
This manuscript is a resubmission of an earlier submission. The following is a list of the peer review reports and author responses from that submission.
Round 1
Reviewer 1 Report
The authors aim to examine dietary disparities in adolescents in Belgium based on their migration strata. The questions investigated in this paper are interesting and important, yet there are several important issues that need to be addressed.
1. In the analysis, all outcome variables are categories into 2-3 categories. Yet such categorization seems to be arbitrary. I’d like to see more justification on that (e.g., why fruit and vegetable consumption have three categories “> once a day,” “5-7 days per week,” :<5 days per week,” whereas fish consumption only has two categories “>= 2 days per week,” and “<2 days per week”?)
2. I like the way how the regression results are presented. But I do think it is important to see the actual numbers in tables for each model (as well as other information such as model fit and sample size). I suggest include the tables in an additional supplementary file.
3. Participation rate is relatively low, especially for French speaking schools (21.5%). This may influence the generalizability of the result. This limitation needs to be addressed/discussed.
Author Response
1. In the analysis, all outcome variables are categories into 2-3 categories. Yet such categorization seems to be arbitrary. I’d like to see more justification on that (e.g., why fruit and vegetable consumption have three categories “> once a day,” “5-7 days per week,” :<5 days per week,” whereas fish consumption only has two categories “>= 2 days per week,” and “<2 days per week”?)
As explained lines 154-155, all food consumption variables were first divided into three categories. This was done in order to have (i) a category that corresponded as closely as possible to the Belgian nutritional policies, (ii) a category further removed from them, and (iii) an intermediate category.
Especially, three categories were first defined for fish (“≥twice a week”, “once a week”, “<once a week”), dairy products (“>once a day”, “once a day”, “<once a day”) and chips and fries (“≤once week”, “2-6 days per week”, “≥once a day”) consumptions. Results of the intermediate categories did not provide additional information, i.e. they are always similar to one of the two others. Therefore, the three categories were reduced to two for these food groups. Contrariwise, the intermediate category of fruit, vegetables, and sugar-sweetened beverages consumptions provided specific information different from the two extreme categories (lines 434-442).
Regarding this point, we have complemented the manuscript lines 155-158.
2. I like the way how the regression results are presented. But I do think it is important to see the actual numbers in tables for each model (as well as other information such as model fit and sample size). I suggest include the tables in an additional supplementary file.
We have added the tables as supplementary files.
3. Participation rate is relatively low, especially for French speaking schools (21.5%). This may influence the generalizability of the result. This limitation needs to be addressed/discussed.
We acknowledge that the participation rate was relatively low, and that the generalizability should be interpreted cautiously.
We have modified the manuscript lines 460-461.
Reviewer 2 Report
The manuscript entitled “Socioeconomic disparities in diet vary according to migration status among adolescents in Belgium” presents interesting issue, but it requires really important corrections before being published
Authors stratified immigrants by various factors (e.g. working status of parents) but not by their native country, while it must be supposed that it is the crucial factor creating dietary habits. It cannot be denied that immigrants from e.g. Asian and African countries may be characterized by various dietary habits, so they should not be analyzed combined.
Other major problems:
1. Authors included both children (e.g. aged 10) and adults (aged 19). I supposed, that Authors based on the definition of adolescents by WHO (indicating the age group of 10-19), but they must realize that for the dietary habits individuals aged 10 and 19 years old are incomparable and should not be analyzed combined
2. The education in Belgium is not compulsory until the age of 19, but until the age of 18, so while recruitment was conducted by schools, the age group of 18-19 years old ones is not representative for the general population but is only representative for population of ones studying in secondary schools (but not for studying at universities or not studying at all).
3. The information that “following advice from school authorities, no written consent was requested” is rather confusing, as Authors state that the study was approved by the ethics review committee of the University Hospital of Ghent, so it should be conducted on the basis of the indicated project (project EC/2013/1145) with no further changes “following advice from school authorities”. It may be an ethical problem.
4. Authors indicated that “in the French-speaking part, the sample was stratified proportionally with the province; in the Dutch speaking part, it was stratified proportionally with the form of education”. It is hard to understand why the French-speaking part was NOT stratified proportionally with the form of education also and the Dutch speaking part was NOT stratified proportionally with the province also. Such inclusion of participants may have contributed to lack of representativeness and result in a serious bias while compared French and Dutch-speaking parts.
5. If the questionnaire is validated, how was it possible that there were a differences in questionnaires applied for French and Dutch-speaking parts (“included a total of 22 food groups (17 in the Dutch part, 18 in the French part, including 13 in common)”)? On the basis of the presented information, it seems that validated was a questionnaire for 22 groups, but the shortened versions applied in the presented study were not validated.
6. Authors should precisely formulate their observations, as they conducted very specific type of analysis and they should not generalize – e.g. “immigrants more frequently consumed both healthy and unhealthy foods” – on the basis of indicated sentence one could suppose that the energy value of their diet was higher, but it was not observed
Abstract:
Authors should briefly formulate the aim of the study (e.g. “The aim of the study was…”)
Introduction:
Authors should present the information more related to the area of their study – e.g. whole first paragraph is not related to the title at all and it is hard to guess why this paragraph is presented.
Hypothesis should not be presented in this section.
Materials and Methods:
Migration status – Authors present the qualification criteria for assessment of the migration status, but not all the possible groups were included – what about adolescents born abroad and whose one parent was not born in Belgium? Was this group not included to study at all?
Results:
Table S1 presented as a supplementary material should be included to main body of the study.
Figure 2 – should be rather presented as a table to be easier to follow
Discussion:
Lines 313-318 – should be removed as reproducing results is not needed
With no information about the native country of immigrants Authors cannot properly discuss the influence of the native and host country eating habits.
Conclusions:
Lines 446-449 – should be removed as reproducing results is not needed
Other:
The manuscript should be formatted according to the instructions for authors (e.g. Figure 1).
Author Response
Authors stratified immigrants by various factors (e.g. working status of parents) but not by their native country, while it must be supposed that it is the crucial factor creating dietary habits. It cannot be denied that immigrants from e.g. Asian and African countries may be characterized by various dietary habits, so they should not be analyzed combined.
Indeed, native country of immigrants, or at least continental regions, would have been interesting to address. However, the relatively small sample size of first-generation immigrants - and consequently of some sociodemographic characteristics - is a considerable limit. For the second-generation immigrants, grouping by geographical region/continent would not overcome very high heterogenous groups, making interpretation still difficult. We agree that it is a limitation of our work.
We now discuss this point lines 446-448.
Other major problems:
1. Authors included both children (e.g. aged 10) and adults (aged 19). I supposed, that Authors based on the definition of adolescents by WHO (indicating the age group of 10-19), but they must realize that for the dietary habits individuals aged 10 and 19 years old are incomparable and should not be analyzed combined
According to the recent work of SM Sawyer (Sawyer, S.M.; et al. The age of adolescence. Lancet Child Adolesc Health 2018,2, 223-228), adolescence corresponds to people aged of 10-24. The 19-year-old adolescents’ lifestyle is probably closer to younger than to adults: most of them (all of them in our study) are still at school and do not have a professional life yet. However, the dietary habits of 19-year-old adolescents might be different to those of 10-year-old adolescents, a reason why we divided age into several groups and adjusted for it.
The manuscript has not been modified.
2. The education in Belgium is not compulsory until the age of 19, but until the age of 18, so while recruitment was conducted by schools, the age group of 18-19 years old ones is not representative for the general population but is only representative for population of ones studying in secondary schools (but not for studying at universities or not studying at all).
Indeed, 19-year-old adolescents are representative of the adolescents attending a secondary school but not of the general population of the same age.
We have made clearer throughout the manuscript that our study concerned school adolescents. In addition, we have modified the manuscript lines 478-460.
3. The information that “following advice from school authorities, no written consent was requested” is rather confusing, as Authors state that the study was approved by the ethics review committee of the University Hospital of Ghent, so it should be conducted on the basis of the indicated project (project EC/2013/1145) with no further changes “following advice from school authorities”. It may be an ethical problem.
The studies in French-speaking schools and in Dutch-speaking schools were independently designed and carried out by two different teams. For the Dutch-speaking schools, the study was approved by the ethics review committee of the University Hospital of Ghent and the consent form was passive. In the French-speaking part, following the Belgian law of May 7, 2004 and the Opinion n°40 of February 12, 2007 of the Advisory Committee on Bioethics of Belgium, the study was not submitted to an ethics committee. School authorities approved and allowed to carry out the study; no written consent was requested following advice from school authorities. To clarify this issue, we have completed the manuscript lines 99-100.
4. Authors indicated that “in the French-speaking part, the sample was stratified proportionally with the province; in the Dutch speaking part, it was stratified proportionally with the form of education”. It is hard to understand why the French-speaking part was NOT stratified proportionally with the form of education also and the Dutch speaking part was NOT stratified proportionally with the province also. Such inclusion of participants may have contributed to lack of representativeness and result in a serious bias while compared French and Dutch-speaking parts.
As mentioned previously, the two protocols were independently developed, and surveys independently conducted by two different teams. The chosen criteria for the stratification of the surveys were different although both were elaborated in order to reach the representativeness of the target populations. In the Dutch-speaking part, stratification was made on the school networks and on the form of education. In the French-speaking part, in addition to the school networks, stratification was made on the provinces. Despite the common hypothesis regarding the representativeness of the final samples, a larger generalization should be interpreted cautiously. Nevertheless, we feel that having grouped the two databases is an important strength of our study since most parts of the protocols are fully comparable.
We have modified the manuscript lines 415-418.
5. If the questionnaire is validated, how was it possible that there were a differences in questionnaires applied for French and Dutch-speaking parts (“included a total of 22 food groups (17 in the Dutch part, 18 in the French part, including 13 in common)”)? On the basis of the presented information, it seems that validated was a questionnaire for 22 groups, but the shortened versions applied in the presented study were not validated.
The HBSC food frequency questionnaire (FFQ) included mandatory items (such as fruit, vegetables…) and optional items (like dairy, bread…). Each region or country made choice among the optional item; this is why the number of items could be different between regions or countries.
The HBSC FFQ has been “validated” by comparing the frequency consumption reported with the FFQ with the frequency consumption reported with a seven-day record (Vereecken et al., 2003; Vereecken et al., 2008). These studies proceeded food item by food item (they did not address nutritional intakes overall for instance). Therefore, there should be no downside to not use all the same items than those in the validated HBSC FFQ.
However, we keep in mind that consumption of certain foods could be overestimated through the FFQ. In addition, the “fish” item was not considered in these studies. Nevertheless, the HBSC FFQ shows reliability for “ranking subjects according to consumption of the individual food items included” as authors stated.
Vereecken, C.A.; Maes, L. A Belgian study on the reliability and relative validity of the Health Behaviour in School-Aged Children food-frequency questionnaire. Public Health Nutr 2003, 6, 581–588.
Vereecken, C.A.; Rossi, S.; Giacchi, M.V.; Maes, L. Comparison of a short food-frequency questionnaire and derived indices with a seven-day diet record in Belgian and Italian children. Int J Public Health 2008, 53, 297–305
We have modified the manuscript lines 424-427.
6. Authors should precisely formulate their observations, as they conducted very specific type of analysis and they should not generalize – e.g. “immigrants more frequently consumed both healthy and unhealthy foods” – on the basis of indicated sentence one could suppose that the energy value of their diet was higher, but it was not observed
Indeed, we did not have other information than the frequency, therefore, we cannot conclude about the energy value or food quantities. Throughout the manuscript, we only stated the frequency of food consumptions. A more frequent consumption does not necessarily imply, nor rules out, a higher food amount or a higher energy intake.
To avoid misunderstanding, we have modified the manuscript lines 427-429.
7. Abstract:
a. Authors should briefly formulate the aim of the study (e.g. “The aim of the study was…”)
We have modified the abstract lines 21-23 accordingly.
8. Introduction
a. Authors should present the information more related to the area of their study – e.g. whole first paragraph is not related to the title at all and it is hard to guess why this paragraph is presented.
The first paragraph was developed to explain why we wanted to address dietary disparities since part of them could explain social inequalities in non-communicable diseases. As requested, we have shortened the section lines 43-47 to focus more on diet than on non-communicable diseases.
b. Hypothesis should not be presented in this section.
According to the ‘Instructions for Authors’ mentioning that introduction should include “hypotheses being tested”, the manuscript has not been modified.
9. Materials and Methods:
a. Migration status – Authors present the qualification criteria for assessment of the migration status, but not all the possible groups were included – what about adolescents born abroad and whose one parent was not born in Belgium? Was this group not included to study at all?
Wording actually was confusing: adolescents born abroad and whose one parent was not born in Belgium were considered to be first-generation immigrants.
We have modified the manuscript lines 134-135.
10. Results
a. Table S1 presented as a supplementary material should be included to main body of the study.
Table S1 has been included to the main body of the study lines 195-199.
b. Figure 2 – should be rather presented as a table to be easier to follow
To our opinion, presenting the results in a figure is the best way to visualize the migration gradient and percentages are available. A table would not facilitate the reading. Therefore, the manuscript has not been modified.
11. Discussion
a. Lines 313-318 – should be removed as reproducing results is not needed
We believe that, given the amount of results and their complexity, a global overview of the results was necessary. We have slightly reduced this paragraph (lines 318-319) in order to only state unreproduced and non-obvious global results.
b. With no information about the native country of immigrants Authors cannot properly discuss the influence of the native and host country eating habits.
Indeed, the country of origin could help better interpret the differences in socioeconomic and sociodemographic disparities between immigrants and natives. This is a limitation of our work. Nevertheless, it does not prevent from making more global assumptions, especially on the influence of the host country, which is the same for every immigrant.
We have modified the manuscript lines 446-450.
12. Conclusions
a. Lines 446-449 – should be removed as reproducing results is not needed
We feel that the few reproduced results in this section are useful for developing the conclusions.
Therefore, the manuscript has not been modified.
13. Other
a. The manuscript should be formatted according to the instructions for authors (e.g. Figure 1).
We have used the template and the styles included from Nutrients to format the manuscript. Following the instructions for authors, figures have been provided in a single zip archive, have been inserted into the main text close to their first citation, have been numbered following their number of appearance and had a short explanatory title and caption.
Round 2
Reviewer 1 Report
The authors are responsive to my comments. I don't have further comments on the revised version of the paper.
Reviewer 2 Report
The manuscript entitled “Socioeconomic disparities in diet vary according to migration status among adolescents in Belgium” presents interesting issue, but it requires really important corrections.
Authors indicated in their title “migration status”, but they did not assess the most important factor associated with this status – namely native country of immigrants. In spite of the fact that Authors stratified immigrants by various factors, they did not analyse this crucial factor creating dietary habits. It cannot be denied that immigrants from e.g. European, Asian and African countries may be characterized by various dietary habits, so they should not be analyzed combined.
Authors explained in their response letter that they did not assess native country of immigrants, as they stated that they had relatively small sample size and heterogenous groups, making interpretation difficult. However, it is not an explanation. Moreover, the fact that there is a small sample size and heterogenous groups influence all the variables, not only the native country of immigrants. At the same time, it is an important bias for all the analysis.
And it is still not known why Authors did not analyse the most important variable to justify indicating “migration status” in their title.
Other major problems:
1. Authors included both children (e.g. aged 10) and adults (aged 19). I supposed, that Authors based on the definition of adolescents by WHO (indicating the age group of 10-19), but they must realize that for the dietary habits individuals aged 10 and 19 years old are incomparable and should not be analyzed combined – Authors should discuss it and indicate as a limitation of the study
2. On the basis of the presented information, it seems that Authors applied food frequency questionnaire that was not validated. Validated was a questionnaire for 22 food product groups, but the shortened versions applied in the presented study (with 17 and 18 food product groups) were not validated. It is a methodological problem, as in general it is required to validate the modified versions of the FFQ, even if they are based on a (previously validated) long version of the questionnaire.
3. Authors should precisely formulate their observations, as they conducted very specific type of analysis and they should not generalize – e.g. “immigrants more frequently consumed both healthy and unhealthy foods” (lines 27-28) – on the basis of indicated sentence one could suppose that the energy value of their diet was higher, but it was not observed
Results:
Figure 2 – should be rather presented as a table to be easier to follow
Discussion:
With no information about the native country of immigrants Authors cannot properly discuss the influence of the native and host country eating habits.
Conclusions:
Reproducing results is not needed
Other:
The manuscript should be formatted according to the instructions for authors (e.g. Figure 1 – font size).